# Anticoagulation Therapy and Severe Traumatic Brain Injury: A Retrospective Cohort Study on Clinical Outcomes Using TriNetX

**DOI:** 10.3390/jcm14134510

**Published:** 2025-06-25

**Authors:** Spencer Rasmussen, Kamal Shaik, Clayton Rawson, Ammar Saloum, Rudy Rahme, Michael Karsy

**Affiliations:** 1Department of Neurosurgery, Drexel University College of Medicine, Philadelphia, PA 19104, USA; str62@drexel.edu (S.R.); kas685@drexel.edu (K.S.); 2Medical College, Noorda College of Osteopathic Medicine, Orem, UT 84606, USA; do27.cdrawson@noordacom.org; 3College of Human Medicine, Michigan State University, Grand Rapids, MI 49503, USA; saloumam@msu.edu; 4Global Neurosciences Institute, Upland, PA 19013, USA; rr999@drexel.edu; 5Department of Neurosurgery, University of Michigan Medical School, Ann Arbor, MI 48109, USA

**Keywords:** severe traumatic brain injury (sTBI), direct oral anticoagulants (DOACs), vitamin K antagonists (VKAs), anticoagulation therapy, TriNetX

## Abstract

**Background**: Traumatic brain injury (TBI) is a leading cause of mortality and disability, particularly in patients on anticoagulation therapy. While anticoagulants are linked to higher TBI mortality, the specific impact of direct oral anticoagulants (DOACs) and vitamin K antagonists (VKAs) on severe TBI (sTBI) outcomes remains unclear, especially in light of newer reversal agents. Therefore, this study evaluates long-term mortality and complication risks associated with pre-injury use of DOACs and VKAs in sTBI patients from a large, real-world cohort. **Methods**: A retrospective cohort study was conducted using the TriNetX global research network, identifying patients with sTBI between 2016 and 2022. Patients were grouped based on pre-injury anticoagulant use: DOAC, VKA, or none. Propensity score matching was performed, adjusting for age, comorbidities, and baseline characteristics. The primary outcome was all-cause mortality at 1-, 3-, 6-, and 12-months post-injury. Secondary outcomes included hospital and surgical complications up to 30 days post-injury. **Results**: A total of 40,563 patients met the inclusion criteria. At all time intervals, no significant mortality differences were found between the PSM-matched groups. **Conclusions**: In patients with sTBI, pre-injury DOAC or VKA use was not associated with increased short- or long-term mortality. These findings suggest that, with current perioperative practices, anticoagulation can be managed without adversely affecting outcomes.

## 1. Introduction

Traumatic brain injury (TBI) is a significant public health concern, often resulting in long-term disabilities or death. It is one of the leading causes of morbidity and mortality worldwide, with millions of individuals affected annually [1,2]. The severity of TBI can range from mild concussions to severe brain damage requiring intensive medical intervention [3,4,5]. Severe TBI (sTBI) is particularly concerning, as it is associated with high rates of neurological impairment, prolonged hospitalization, and an increased burden on healthcare systems. Advances in neurocritical care have improved survival rates; however, optimizing and establishing treatment guidelines remains a major challenge, particularly in specific patient populations such as those receiving anticoagulant therapy [4,6,7].

Increased use of anticoagulant therapies, such as direct oral anticoagulants (DOACs) or vitamin K antagonists (VKAs), has been observed for the management of thromboembolic conditions (e.g., atrial fibrillation, venous thromboembolism, mechanical heart valves) despite an increased risk of intracranial bleeding. Patients on anticoagulants who sustain sTBIs frequently experience worse outcomes, including elevated mortality rates, emphasizing the urgent need for targeted strategies to optimize care [1]. The anticoagulation-induced impairment of hemostasis presents a significant concern in trauma patients, as even minor injuries may lead to catastrophic hemorrhagic complications [4,8]. While emerging evidence suggests that DOACs may pose a lower bleeding risk compared to VKAs, robust clinical data evaluating their impact of different anticoagulants on sTBI outcomes are lacking. The reversal of anticoagulation presents a complex clinical dilemma, as the risk of thrombotic events must be balanced against the need for hemorrhage control. Novel therapies include non-specific three- or four-factor prothrombin complex concentrate, idarucizumab for dabigatran, and Andexanet alfa for apixaban and rivaroxaban [9,10,11,12].

Despite the growing prevalence of anticoagulant use, there remains a critical gap in understanding how DOACs and VKAs influence outcomes specifically in patients with sTBIs. To address this gap, we analyzed the TriNetX global health research network to evaluate mortality and complication rates in sTBI patients according to pre-injury anticoagulant status, providing insights to inform future management.

## 2. Materials and Methods

The TriNetX Research Network database was retrospectively studied on 29 April 2025. The TriNetX database does not involve identifiable patient information and is subsequently exempt from Institutional Review Board review and approval. TriNetX (Cambridge, MA, USA) is a global research network encompassing data from over 170 healthcare organizations and more than 400 million patients. It contains de-identified aggregate patient information covering procedures, diagnoses, medications, vitals, genomics, and demographics. Healthcare organizations (HCOs) involved in the TriNetX network contribute healthcare data in de-identified, pseudo-anonymized, or limited data set formats, following local privacy regulations. These HCOs authorize the usage of this data for research purposes on the TriNetX platform. In return for providing data, HCOs incur no financial expenses and gain access to data query tools, analytics, visualization capabilities, and the necessary hardware for software execution. The de-identification process conforms to HIPAA Privacy Rule standards, as verified by a qualified expert, meeting the requirements of Section §164.514(b)(1), and ensuring HIPAA compliance [13].

Patients of all ages diagnosed with an sTBI between January 2016 and December 2023 were included in the study. Classification of sTBI utilized the International Classification of Diseases 10th Edition (ICD-10) diagnosis codes and Logical Observation Identifiers Names and Codes (LOINC). Due to institutional variability in defining sTBI, the NIH coding guidance for TBI was used to try and capture a standardized cohort of patients with a TBI [14]. The classification of sTBI required a patient to have a GCS of 3-8 (ICD-10 code R40.243 OR LOINC code 9269-2) in addition to one of the following: intracranial injury (ICD-10 code S06), fracture of vault of skull (ICD-10 code S02.0), fracture of base of skull (ICD-10 code S02.1), fractures of other specified skull and facial bones (ICD-10 S02.8), other and unspecified injuries of head (ICD-10 code S09), concussion (ICD-10 code S06.0), traumatic cerebral edema (ICD-10 code S06.1), diffuse traumatic brain injury (ICD-10 code S06.2), focal traumatic brain injury (ICD-10 code S06.3), or other amnesia (ICD-10 R41.3). This combined diagnosis was used as the index event.

Pre-TBI anticoagulant use was identified using Veterans Affairs (VAs) and RxNorm medication codes for DOACs (rivaroxaban, edoxaban, betrixaban, dabigatran etexilate, or apixaban) and VKAs (warfarin), defined as prescriptions occurring at least 1 day prior to the sTBI index event. Patient characteristics were then compared across three groups: DOAC + sTBI, VKA + sTBI, and a control group with sTBI excluding anticoagulant use. Patient selection criteria are outlined in Figure 1.

Selected demographic variables and comorbid diagnoses were required to be documented on or before the index event to be included in baseline comparisons and propensity score matching. This included age, sex, race, disease of the respiratory system (J00-J99), disease of the cardiovascular system (I00-I99), diabetes mellitus (E08-E13), acute kidney injury and chronic kidney disease (N17-N19), overweight, obesity, and hyperalimentation (E65-E68).

The primary outcome under analysis was mortality risk at 1 month, 3 months, 6 months, and 12 months post-sTBI. Secondary outcomes included urinary tract infection (UTI) (N10, N39.0, and N30), deep vein thrombosis (DVT) (I82.40-I82.46, I82.49, I82.4Y, I82.4Z), pneumonia (J12-J18), stroke (I63), myocardial infarction (I21), gastrointestinal bleed (K92.2), anticoagulant reversal agent administration (Warfarin—Prothrombin complex concentrate and Vitamin K; DOAC—andexanet alfa and idarucizumab), nontraumatic subdural hemorrhage (I62.0), nontraumatic subarachnoid hemorrhage (I60) as well as hospital- (short stay) and surgical- (tracheostomy, PEG tube, craniectomy/craniotomy, seizure, and pulmonary embolism) related outcomes at 1 month post sTBI.

All statistical analyses were performed using the built-in TriNetX analysis platform. A *p*-value of less than 0.05 was used to indicate statistical significance. Cohort demographic and comorbid differences were assessed using two-sided independent sample t-tests and chi-squared tests. Propensity score matching utilizes a greedy nearest neighbor algorithm based on characteristics and comorbid conditions to minimize selection bias. Risk ratios (RRs) with 95% confidence intervals (CIs) were calculated to compare primary and secondary outcomes between cohorts within TriNetX, and then summary statistics were manually extracted and graphically visualized using Microsoft Excel (Microsoft Corporation, Redmond, WA, USA; version 16.98). Microsoft PowerPoint (Microsoft Corporation, Redmond, WA, USA; version 16.98) was used to generate Figure 1. ChatGPT 4.5 (OpenAI, San Francisco, CA, USA), a large language model, was used to assist with editing and refining portions of the manuscript draft. All content was reviewed and finalized by the authors.

## 3. Results

### 3.1. Patient Characteristics

A total of 40,563 patients were identified with a diagnosis of sTBI, of whom 36,724 had no anticoagulant use and 3839 had some anticoagulant use (VKA or DOAC) prior to trauma. The DOAC group (*n* = 1942; median age 70 ± 16 years; 1073 male [55.25%], 811 female [41.76%]; 1219 [62.77%] White, 358 [18.44%] Black, 60 [3.09%] Asian) and the VKA group (*n* = 1897 median age 69 ± 17 years; 1056 male [55.67%], 787 female [41.49%]; 1216 [64.10%] White, 347 [18.29%] Black, 48 [2.53%] Asian) were similar in terms of demographics, comorbidities, and BMI. Propensity-score matched patient characteristics yielded treatment and control groups of equal size. From the control group (*n* = 36,724), 1942 patients were matched to the DOAC group, and 1897 patients were matched to the VKA group. No significant characteristic differences were observed in the matched groups (Table 1).

### 3.2. Primary Outcome

No worsened mortality was seen in the DOAC group (1 month: RR = 1.03, 95% CI [0.94, 1.13], *p* = 0.50; 3 months: RR = 1.04, 95% CI [0.96, 1.13], *p* = 0.29; 6 months: RR = 1.00, 95% CI [0.93, 1.08], *p* = 0.93; 12 months: RR = 1.01, 95% CI [0.94, 1.08], *p* = 0.85). Similarly, the VKA group showed no worsened mortality (1 month: RR = 0.98, 95% CI [0.90, 1.06], *p* = 0.57; 3 months: RR = 0.94, 95% CI [0.87, 1.02], *p* = 0.12; 6 months: RR = 0.94, 95% CI [0.88, 1.01], *p* = 0.11; 12 months: RR = 0.95, 95% CI [0.89, 1.02], *p* = 0.17) (Table 2 and Table 3).

### 3.3. Secondary Outcomes

Secondary outcomes of post-injury complications, hospital complications, and post-surgical complications were statistically insignificant in the DOAC and VKA groups when compared to the matched control groups. A significantly greater proportion of patients in the VKA group received anticoagulant reversal agents within 30 days post-injury (RR = 0.77, 95% CI [0.63–0.93], *p* = 0.0069) (Table 4 and Table 5).

## 4. Discussion

The literature on mortality outcomes in TBI patients using anticoagulants remains complex and, at times, contradictory. While several studies report increased mortality in anticoagulated patients, particularly those using VKAs, others show no difference—or even decreased mortality—in those receiving DOACs [15,16,17]. In our multicenter cohort of patients with sTBI, we observed no significant difference in all-cause mortality at 1, 3, 6, or 12 months between those on preinjury DOACs, VKAs, or no anticoagulation, after propensity matching. These results suggest that anticoagulation status alone does not independently predict mortality in sTBI patients when accounting for baseline covariates.

Several systematic reviews and observational studies have found elevated mortality among anticoagulated patients [15,16,17]. Lim et al. (2021), for example, identified a significantly increased mortality risk in anticoagulated TBI patients, while Posti et al. (2022) reported that patients on VKAs had higher short-term mortality rates than non-anticoagulated individuals [18,19]. Similarly, Pedro et al. (2024) and Scotti et al. (2019) noted that warfarin use was associated with worse mortality outcomes and an increased risk of intracranial hemorrhage (ICH), whereas DOAC users tended to fare better, though not uniformly so [20,21]. In contrast to these findings, our study did not demonstrate increased mortality in either VKA or DOAC groups, suggesting that in the setting of severe TBI, injury burden and matched patient cohorts may diminish the observable impact of anticoagulation on survival. Differences in the study population, such as focusing on the elderly or criteria for how a TBI was defined, may have had an impact on the differences in the results.

Other investigations have not found significant differences in mortality outcomes. Studies by Rønning et al. (2021), Nederpelt et al. (2020), and Kobayashi et al. (2017) did not observe higher mortality in anticoagulated patients compared to controls, aligning more closely with our results [22,23,24]. The consistency between our findings and these studies may reinforce the notion that contemporary trauma systems and TBI management protocols can mitigate risks previously attributed to anticoagulation.

The meta-analysis by Pan and Hu (2024), interestingly, describes an “anticoagulant paradox,” in which anticoagulated patients showed lower mortality compared to controls—a finding they hypothesized could reflect protective effects of anticoagulation or more attentive clinical management [17]. While intriguing, we did not observe this paradox in our cohort. The mortality rates in our DOAC and VKA groups were similar to those in the non-anticoagulated cohort. This difference may stem from our focus on severe TBI, in contrast to their broader inclusion of injury severities, where survival margins are more flexible and susceptible to care-level differences. Furthermore, the use of a meta-analysis can introduce variations in results due to interstudy variability.

Methodological differences across the literature likely contribute to divergent findings. Differences in TBI definitions, anticoagulant exposure classification, follow-up durations, and data sources complicate direct comparisons. Our study’s use of strict inclusion criteria for sTBI, a multicenter real-world data platform, and robust propensity score matching provides a distinct lens through which these mortality outcomes can be examined and contextualized within this broader evidence base.

Anticoagulant reversal in TBI management remains a subject of ongoing debate. While reversal strategies are commonly employed to mitigate bleeding risks, emerging evidence suggests that their benefits may vary depending on injury severity, patient characteristics, and the type of anticoagulant involved. For warfarin-associated ICH, studies support the use of prothrombin complex concentrates (PCCs) to reduce hematoma expansion and improve survival [25]. In DOAC-associated ICH, specific reversal agents such as andexanet alfa and idarucizumab have demonstrated high rates of effective hemostasis, though concerns persist regarding thromboembolic risks and unclear effects on functional outcomes [26,27,28,29]. In our study, reversal agent administration was captured as a binary outcome and did not evaluate dosage or timing post-injury, as such granularity was beyond the scope of our study’s focus on pre-injury anticoagulant use. Therefore, these results suggest that patients on warfarin prior to injury were significantly more likely to receive a reversal agent, reflecting contemporary practice patterns and institutional protocols.

Several of the strengths of our study support the overall findings. Unlike studies that mix mild, moderate, and severe TBI, our exclusive focus on sTBI may reduce variability and confounding from injury severity. Additionally, our use of propensity score matching allowed for high-precision control of demographic and comorbidity confounders, potentially isolating the true effect of anticoagulant exposure. The large real-world data set accessed through TriNetX provided robust patient matching and long-term follow-up capabilities, enhancing the reliability of longitudinal mortality assessments.

Our study is not without limitations. First, the retrospective design introduces the potential for residual confounding despite propensity score matching. The retrospective nature of the study may lead to incomplete or inaccurate data from non-standardized records and selection bias that may reduce the representativeness of the general population. While matching enhances comparability, it may inadvertently include or exclude patients who do not align with the original target population based on the covariates that were selected, thus making it difficult to account for all the potential confounding variables. Second, granular data on intracranial hemorrhage volumes, surgical intervention timing, and reversal agent dosage and timing were not available in the outcomes analyses. The inability to take the analysis a step further may prevent more concrete associations from being made regarding anticoagulation and patient outcomes. Third, although TriNetX provides robust population-level data, institutional variability in trauma management and data entry may limit the generalizability of our findings. These differences that can occur at patient presentation, all the way to discharge, can introduce significant bias, which may alter the findings associated with our study. Finally, we did not examine functional outcomes or quality of life measures, which are critical in understanding the full burden of sTBI. Without this information, unfortunately, there is no concrete way to determine the full impact that pre-injury anticoagulation can have on these patients while using this data set. This limitation stems from the fact that the TriNetX database does not provide measurable or standardized data points for assessing these outcomes.

## 5. Conclusions

In this large multicenter cohort study of patients with sTBI, we found no significant differences in short- or long-term mortality between those on preinjury direct oral anticoagulants (DOACs), vitamin K antagonists (VKAs), or no anticoagulation. These findings differ from studies reporting increased mortality among anticoagulated patients—particularly VKA users—but are consistent with the literature, suggesting comparable outcomes when patient demographics, comorbidities, and injury severity are adequately controlled. We also observed significantly higher rates of reversal agent use among VKA users, reflecting current clinical practices, though the clinical impact of reversal remains uncertain. Taken together, our results suggest that anticoagulation status alone may not independently determine mortality risk in sTBI when patients receive timely and appropriate trauma care. As this study is aimed at generating hypotheses for future research. Further studies are warranted to clarify the role of anticoagulant reversal, the impact of other unique comorbidities or medications, and to identify patient subgroups that may benefit most from targeted interventions.

## Data Availability

Restrictions apply to the availability of these data. Data were obtained from TriNetX, LLC, and are available from the authors with the permission of TriNetX, LLC.

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
