# Peer review of "Anticoagulation Therapy and Severe Traumatic Brain Injury: A Retrospective Cohort Study on Clinical Outcomes Using TriNetX"

_jcm, 2025, doi:10.3390/jcm14134510_

Round 1
Reviewer 1 Report
Comments and Suggestions for Authors
Dear authors,
thank you for this paper which brings up important clinical results considering the use of anticoagulant therapy in cases of severe brain trauma, and analysis of the outcome of those patients considering DOAC's and/or VKA as well. In spite of being retrospective, the huge number of included patients provides a respectful level of statistical certainty considering the results.
I would kindly ask you just for a few explanations:
- inclusion criteria of use of VKA 1 day prior to sTBI: 1 day use of VKA does not provide a therapeutic level of anticoagulation in a high percentage of patients, on the other hand, 1 day use of DOAC's does provide therapeutic level of anticoagulation in the most of the patients. Could did fact contaminate the results (no difference in outcome between the groups)? Could you please explain.
- Anticoagulant reversal has been done mostly in VKA group. Do you have any data about the indications for anticoagulant reversal? Has this fact been analysed considering the outcome (better outcome in "reversal" group)? Could you please comment on this
Thank you for your answers and congratulation for this huge work
Reviewer 2 Report
Comments and Suggestions for Authors 1. In the line 73 - 74 the author mentions that the use of data collectionfrom participants was approved by the Institutional Review Board approval
through the Drexel College of Medicine. However, in line 252, the author
mentions a Institutional Review Board Statement: "Ethical review and approval
were waived for this study". Please review this contradiction. In general,
although participants are exempt from signing the ICF in studies that use
the TriNetX Platform database due to robust data anonymization, it is more
common and appropriate do an institutional ethics committee review to
provide approval for the use of the data.
It is important include and mention on the limitations of the study other
concerns due TriNetX Platform database use. For example, although the
TriNetX database is well known, and corresponds to a database with
standardized and homogenized information, it does not correspond to the
collection of primary data, focused and carried out with objective accuracy
in research. They can be loaded with bias due to real medical context.
For example, the diagnosis of TB in the included participants was carried
out based on the description of the ICD-10 collected in the medical records
and may have inaccuracies arising from the care team under pressure or over
loaded with day-to-day tasks. Please expands the discussion regarding
limitations by also mentioning the limitations of the chosen data source
method.
Since the correlation between variables is extremely important to interpretation
of the results, I suggest that the editor carry out an evaluation with a
statistics specialist.
Reviewer 3 Report
Comments and Suggestions for Authors
Content suggestions:
- For the completness, can the Authors specify statistical methods used in the study ?
- Can the Authors indicate whether patients included in the study took acetylsalicylic acid and its derivatives due to their comorbidities ? I suppose that such patinets were excluded from the study.
- Because of interesting findings related to the use of reversal agents, I would like to kindly ask the Authors to perform deeper analysis to correlate between the effectiveness and dose of reversal agent + time of its administration after TBI.
Formal suggestions:
- The Table 4 occurs two times and Table 5 is not present at all. I suppose this is just a matter of nsaming the last table. Please, clarify it.
